# Observe before Generate: Emotion-Cause aware Video Caption for Multimodal Emotion Cause Generation in Conversations

## ABSTRACT

Emotion cause analysis has attracted increasing attention in recent years. Extensive research has been dedicated to multimodal emotion recognition in conversations. However, the integration of multimodal information with emotion cause remains underexplored. Existing studies merely extract utterances or spans from conversations as cause evidence, which may not be concise and clear enough, especially the lack of explicit descriptions of other modalities, making it difficult to intuitively understand the causes. To address these limitations, we introduce a new task named Multimodal Emotion Cause Generation in Conversations (MECGC), which aims to generate an abstractive summary describing the causes that trigger the given emotion based on the multimodal context of conversations. We accordingly construct a dataset named ECGF that contains 1,374 conversations and 7,690 emotion instances from TV series. We further develop a generative framework that first generates emotion-cause aware video captions (Observe) and then facilitates the generation of emotion causes (Generate). The captioning model is trained with examples synthesized by a Multimodal Large Language Model (MLLM). Experimental results demonstrate the effectiveness of our framework and the significance of multimodal information for emotion cause analysis.

## CCS CONCEPTS

• **Information systems** → **Sentiment analysis**; • **Computing methodologies** → **Natural language processing**.

## KEYWORDS

multimodal emotion cause generation, emotion cause analysis in conversations, video captioning

## 1 INTRODUCTION

Emotion is an inherent attribute of humans, reflecting psychological responses to internal and external stimuli, encompassing various types such as happiness and anger, thereby influencing our thinking and behavioral patterns. Therefore, delving into the emotion cause not only aids in enhancing self-awareness and promoting mental health, but also strengthens empathic abilities and optimizes interpersonal relationships.

The pioneer studies on emotion cause analysis primarily focused on scenarios such as news articles, fiction stories, and social media

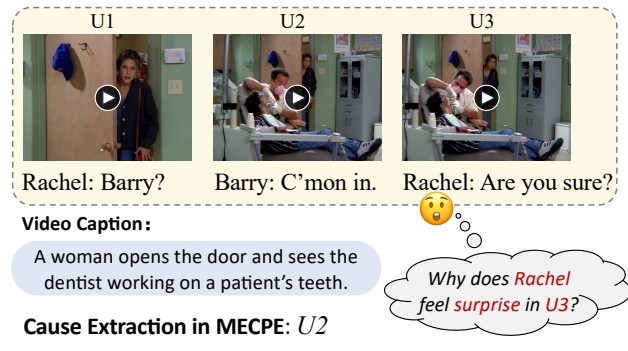

**Figure 1: An example showing the difference between Emotion Cause Extraction and our Emotion Cause Generation task in a multimodal conversation scenario. In previous work, "U2" was annotated for emotion cause extraction that aims to predict which utterance contains the cause cues. Our MECGC task shifts the focus from extraction to the generation of abstractive emotion causes, and our dataset ECGF is constructed by manually annotating the textual descriptive cause.**

posts [4, 8, 11, 19, 37]. In recent years, conversation, as the main style of human information exchange and emotional expression, is gaining more interest in emotion cause analysis. This helps facilitate understanding and reduce conflicts during conversation. Initially, researchers focused on identifying the causes of emotion in textual conversations [3, 20, 30, 36], ignoring the importance of multimodal information in discovering causes. Until recently, emotion cause analysis in conversation has expanded from textual settings to multimodal settings including audio and vision. Wang et al. [35] introduced the task of Multimodal Emotion Cause Pair Extraction in conversations (MECPE) and built the ECF dataset where emotion causes are annotated at the utterance level.

However, all the aforementioned research regards emotion cause analysis as an extraction problem, either extracting text spans as cause evidence, or predicting which utterance contains the cause cues. As the causes may be reflected in the audio or vision modality, such an extractive manner is not intuitive enough and overlooks audio-visual cues. Figure 1 shows an example of a multimodal conversation. From the video clips, we can see that Rachel opens the door and sees Barry, a dentist, treating a patient. Then she shows *surprise* in *U3* after Barry lets her in. In the ECF data set, the multimodal utterance *U2* is annotated as the cause for Rachel's surprise, which is vague and fails to capture the implicit cause present in the visual scene.

To overcome this issue, in this work, we propose to perform emotion cause analysis in conversations in a generation manner, and introduce a new task named Multimodal Emotion Cause Generation in Conversations (MECGC). Given an emotion utterance, MECGC aims to integrate information from multiple modalities to directly generate a corresponding abstractive cause that summarizes the factors triggering the emotion in the given utterance. As illustrated in Figure 1, for Rachel's surprise emotion, MECGC is expected to generate an abstractive emotion cause: "Barry invites Rachel to come in while he has a patient", incorporating multimodal information from the vision modality showing Barry treating a patient and the text modality "C'mon in". Compared to simply extracting the second utterance (i.e., "U2") as the emotion cause in MECPE, the cause summary in our MECGC task is more intuitive and fine-grained.

Since there was no available dataset for MECGC, we construct a new dataset named ECGF based on the existing ECF dataset, which includes 1374 conversations and 13169 utterances from the famous TV series Friends. For the 7690 utterances that contain emotions, we meticulously annotate the abstractive causes by integrating audio-visual information and conversation texts after observing the scenes.

To address the new MECGC task, we further propose a multimodal pipeline framework named "Observe-before-Generate" (ObG), which first generates the emotion-cause aware video caption and then incorporates the generated caption for emotion cause generation. Specifically, we first leverage the powerful few-shot visual understanding capabilities of a representative multimodal large language model (MLLM) named Gemini [33] to generate emotion-cause aware video captions, filtering out irrelevant visual information in complex scenes and focusing only on the parts related to the given emotion and cause. We then use them as the supervised data to fine-tune a smaller multimodal pre-trained model ECCap. Finally, we integrate the emotion-cause aware visual clues from ECCap into a pre-trained encoder-decoder model, and fine-tune it for emotion cause generation.

We conduct extensive experiments on our dataset over a variety of pre-trained unimodal and multimodal models, including T5, Flan-T5, GPT-3.5, Gemini and Gemini-Pro. Both automatic and human evaluation results demonstrate the superiority of the proposed ObG framework, highlighting the advantages of leveraging emotion-cause aware video captions. Remarkably, when applied to Flan-T5, our framework exhibits significant improvements over Gemini-Pro in emotion cause generation.

The main contributions of our work can be summarized as follows:

- We introduce a task named Multimodal Emotion Cause Generation in Conversations (MECGC), aiming to generate abstractive causes based on multimodal context.
- We construct a dataset named ECGF for the MECGC task by annotating abstractive causes for each emotion based on the existing ECF datset.
- We propose a multimodal pipeline framework named "Observe-before-Generate" (ObG), which first trains an ECCap model with examples synthesized by a MLLM to generate emotion-cause aware video captions and then uses them for emotion cause generation.

## 2 RELATED WORK

### 2.1 Multimodal Emotion Recognition in Conversations

Multimodal Emotion Recognition in Conversations (MERC) constitutes a prevalent research domain within sentiment analysis, which endeavors to ascertain the emotions embedded in each utterance from a pre-defined set, considering the conversation context (encompassing text, audio, and video) alongside speaker information [29]. Most existing studies on the MERC task have engaged various neural networks modeling inter-modal interactions and acquiring contextual information in the conversation, including earlier methods based on gated recurrent unit [13], graph-neural-network-based methods [14, 16, 17, 22], transformer-based methods [5, 24], and methods utilizing pre-trained language models [15, 26].

### 2.2 Emotion Cause Analysis

Emotion Cause Analysis aims to analyzes underlying emotions and the corresponding causes, which has gained increasing interests recently. It comprises of two representative subtasks: Emotion Cause Extraction (ECE) and Emotion Cause Pair Extraction (ECPE). The genesis of the Emotion Cause Extraction task was first introduced by [19] to identify textual segments that elucidate the causes behind the specified emotions. Based on this foundation, Gui et al. [12] expanded the ECE task to clause-level analysis and released a newly constructed Chinese emotion cause dataset. Moreover, since ECE necessitates the pre-annotation of emotions and overlooks the relations between the identification of emotions and their causes, Xia and Ding [37] introduced the ECPE task, which simultaneously discerns and extracts emotions and their corresponding causes embedded within texts. In recent years, researchers have increasingly focused on ECA in conversations [20, 30, 39, 41]. To enhance the comprehension of emotional expressions by all participants in a conversation and to elucidate the reciprocal effects among spoken contributions, Li et al. [21] introduced the ECPEC task, extending the ECPE task from news to conversation scenario. Wang et al. [35] further integrated multimodal information into the task to explore the possible causes in the audio and visual modalities.

### 2.3 Cause Generation

Recent developments in pre-trained models have demonstrated significant advancements in a variety of inference tasks. Numerous studies have leveraged generative pre-trained models for downstream tasks by fine-tuning on task-specific dataset, facilitating the generation of summaries or explanations that are comprehensible to humans. Given that the summarization process is intrinsically related to reasoning, Ghosal et al. [10] modeled causal reasoning in conversations in a unified manner of question answering and multiple choice, utilizing models such as T5 to generate chain-style causal explanations. Moreover, there has been an exploration into the generation of textual emotion causes in recent studies. Riyadh and Shafiq [32] concentrated on the generation of causes for a pre-defined causes within specific sentences, whereas Zhan et al. [38] endeavored to ascertain emotions and encapsulate the causative triggers within social media contexts. Nguyen et al. [27] built a

**Table 1: Comparison between our ECGF and existing datasets for Emotion Cause Analysis**

| Dataset | Language | Modality | Source | Size | Cause |
|---|---|---|---|---|---|
| Emotion-Stimulus [9] | English | T | FrameNet | 2,414 sentences | Extractive |
| ECE Corpus [11] | Chinese | T | SINA city news | 2,105 documents | Extractive |
| NTCIR-13-ECA [8] | English | T | Novel | 2,403 documents | Extractive |
| Weibo-Emotion [4] | Chinese | T | Sina Weibo | 7,000 tweets | Extractive |
| REMAN [18] | English | T | Fiction | 1,720 documents | Extractive |
| GoodNewsEveryone [2] | English | T | News headlines | 5,000 sentences | Extractive |
| RECCON-IE [30] | English | T | Acting records | 665 utterances | Extractive |
| RECCON-DD [30] | English | T | English learning websites | 11,104 utterances | Extractive |
| ConvECPE [23] | English | T | Acting records | 7,433 utterances | Extractive |
| ECF [35] | English | T,A,V | TV series | 13,619 utterances | Extractive |
| COVIDET [38] | English | T | Reddit | 1,883 posts | Abstractive |
| EMO-KNOW [27] | English | T | Twitter | 772,863 tweets | Abstractive |
| **ECGF (Ours)** | English | T,A,V | TV series | 13,619 utterances | Abstractive |

dataset containing tweets where users describe their emotions triggering events, spanning 48 emotion types.

## 3 TASK DEFINITION

In a conversation containing multiple utterances $D = [U_1, \ldots, U_n]$, each utterance corresponds to a speaker and consists of three modalities: text, audio, and vision, denoted as $U_i = [s_i, u_i^t, u_i^a, u_i^v]$. The goal of our task, Multimodal Emotion Cause Generation in Conversations (MECGC), is to generate a text sequence $Y = (y_1, \ldots, y_m)$ describing the cause that triggers the given emotion $e_i$ in the target utterance $U_t$, where $e_i$ is one of Ekman's six basic emotions, including *Anger, Disgust, Fear, Joy, Sadness* and *Surprise* [7]. In this task, the emotion cause is no longer the span or utterance extracted from the conversation [30, 35], but an abstractive summary based on the multimodal context. As shown in Figure 1, given the multimodal conversation and Rachel's *joy* emotion in U3, the output of our MECGC task is the abstractive cause: *Barry invites Rachel to come in while he has a patient.*

## 4 DATASET CONSTRUCTION

### 4.1 Data Source

Given that TV series closely resemble the real world and contain rich emotions and causes, we choose the ECF dataset developed by [35] as our data source, which consists of conversations from the famous American TV series *Friends* and is currently the largest available multimodal emotion cause dataset.

The ECF dataset is built for the task of emotion cause extraction, where emotions and causes are annotated at the utterance level. On this basis, we retain its emotion labels and re-annotate the cause for each emotion by asking annotators to write sentences describing the causes.

### 4.2 Annotation Process

*4.2.1 Annotators.* We employed a total of five annotators who are graduate students majoring in computer science and have adequate knowledge about emotion cause analysis. Two of them were responsible for writing the emotion causes, while the remaining three

served as evaluators and voted to determine whether to keep the cause annotations as the ground truth.

*4.2.2 Annotation Guidelines.* We randomly shuffled the conversations in the ECF dataset and evenly assigned them to two annotators, ensuring that each conversation was annotated by one annotator. Given a conversation, annotators first watched the corresponding video clip from the episode to gain a comprehensive understanding of the context and the audio-visual scenes. For each utterance labeled with an emotion category, the annotators were required to think about "why the speaker shows this emotion" and write the cause based on the entire multimodal conversation. Cause annotations should adhere to the following requirements:

- be written in the third person;
- be clear and concise, consisting of one to three sentences;
- accurately describe the event or opinion that triggers the emotion.

If the original conversation text explicitly expresses the cause, annotators can excerpt sentences from it. Additionally, in rare cases where the causes are latent, i.e., the causes cannot be directly inferred from the limited multimodal conversation, annotators were allowed to speculate on reasonable causes based on the plot of the episode.

*4.2.3 Annotation Quality.* Before formal annotation, we randomly selected 100 conversations as trial data to train and evaluated the annotators. Only annotators who provided satisfactory annotations on the trial data were qualified to annotate the entire dataset. During the annotation process, we regularly reviewed the annotations and discussed any issues encountered, providing the annotators with timely feedback and guidance. Annotators were allowed to modify previous annotations to address misunderstandings or other problems, ensuring the accuracy and consistency of their annotations. After the annotation was completed, three additional evaluators were instructed to check the cause annotation for each emotion and assess its accuracy, voting on "keep" or "discard". If two or three evaluators agreed with the annotation and voted "keep", it will be regarded as the ground truth. About 88.3% of the annotations were unanimously approved by the three evaluators, indicating

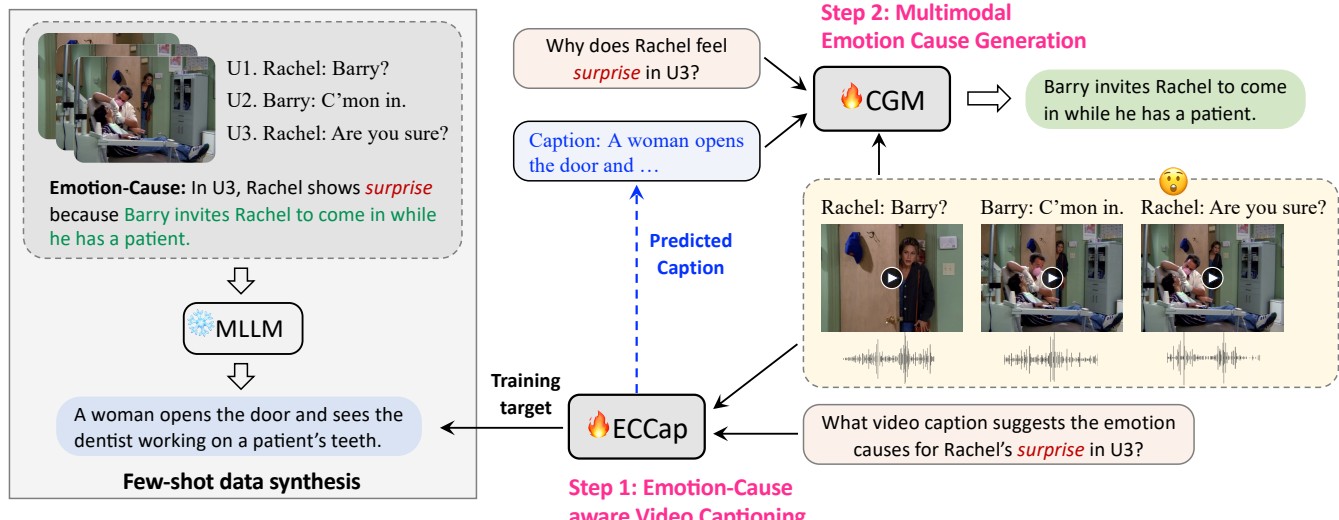

Figure 2: The overview of our proposed multimodal pipeline framework ObG. It consists of two components: Emotion-Cause aware Video Captioning (ECCap) model and Cause Generation Model (CGM). ECCap is trained using supervised datas syntheis by a MLLM, while CGM generate the abstractive cause for the given emotion, utilizing the emotion-cause aware caption generated by ECCap.

Table 2: Basic Statistics of Our ECGF Dataset

| Items | Train | Val | Test | Total |
|---|---|---|---|---|
| # of Conversations | 1,001 | 112 | 261 | 1,374 |
| # of Utterances | 9,966 | 1,087 | 2,566 | 13,619 |
| # of Emotions | 5,577 | 668 | 1,445 | 7,690 |
| Average Cause Length | 10.19 | 10.51 | 10.68 | 10.31 |

satisfactory annotation quality of our dataset. For annotations with at least two "discard" votes, we assigned these instances to another expert to re-annotate the causes based on the feedback from the evaluators and the conversation context.

### 4.3 Dataset Statistics and Analysis

In Table 1, we compare our dataset with the existing datasets for emotion cause analysis, in terms of language, modality involved, data source, size, and cause type. Extractive causes refer to text spans or entire clauses/utterances from the origin instance, while abstractive causes are summarized descriptions of causes. Our dataset is the first multimodal conversational emotion cause dataset that provides abstractive cause annotations.

The basic statistics of our data set are shown in Table 2. We adopt the same data split as in ECF. It can be seen that each conversation has an average of about 10 utterances, and about 55.7% of the utterances are labeled with one of the six basic emotions. Each emotion utterance has a corresponding cause annotation in our dataset, including some (approximately 8.7%) that are not annotated with extractive causes in ECF. Our abstractive causes are concise and average about 10 words.

To compare cause annotations, we measure the similarity between the textual cause spans in ECF and the abstractive causes in our dataset using text generation metrics. Specifically, we calculate the degree of lexical overlap through BLEU-4 [28], METEOR [1] and ROUGE-L [25], with scores of 0.1970, 0.2061, and 0.2714, respectively. We also assess the semantic similarity through BERTScore [40], with an F score of 0.5915. Our annotated causes are manually paraphrased and include words or phrases beyond the conversation text, thereby reducing the n-gram overlap. The cause spans may contain key cause clues, making them somewhat similar to ours in terms of semantics.

## 5 METHODOLOGY

As illustrated in Figure 2, our multimodal pipeline framework ObG consists of two components: Emotion-Cause aware Video Captioning (ECCap) model and Cause Generation Model (CGM). First, we utilize MLLMs to synthesize emotion-cause aware video captions and finetune a pre-trained multimodal model with these captions to acquire ECCap. Then, we apply EACap for emotion-cause aware caption generation, integrate them into the multimodal conversations, and input them into CGM to generate the abstractive cause for the given emotion. Both parts are based on the pre-trained model with encoder-decoder architecture, and integrate three modalities including text, audio and vision.

### 5.1 Emotion-Cause aware Video Captioning

Information from the vision modality often involves complex scenes, characters, and actions. Video modality information related to a specific emotion usually constitutes only a small part of the total information. ECCap aims to filter out visual information that is irrelevant to the emotion cause, retaining only the key visual

**Figure 3: The instruction template for supervised data synthesis using Gemini.**

information directly related to the emotion cause, comprising of three main modules: Few-shot Data Synthesis, Unimodal Feature Extraction, and Model Training.

*5.1.1 Few-shot Data Synthesis.* To empower ECCap with the ability to generate emotion-cause aware captions, while controlling human labor cost, we adopt a strategy of distilling data from MLLM for training ECCap.

Specifically, we adopt Gemini-pro[1], the multimodal version of Gemini [33], as the MLLM for generating synthetic emotion-cause aware video captions. Figure 3 demonstrates the synthesis process for each multimodal sample in ECGF. First, we use the FFmpeg[2] tool to extract key frames from each utterance corresponding video clip as image inputs. Second, we input each utterance's image frames and texts along with annotated emotions and causes into Gemini. Considering the input limitation of up to 16 images for Gemini, we adopt a sliding window method to shorten the input, i.e., selecting the emotional utterance and its previous three utterances as the context, where most visual clues are likely to appear. Finally, we instruct Gemini to focus on key visual clues related to the emotion cause to describe these images. We also adopt in-context learning to enhance the quality of synthesized data generated by Gemini, for which we manually write three emotion-cause aware caption samples as Gemini's demonstration.

---

[1]https://ai.google.dev/
[2]https://www.ffmpeg.org

*5.1.2 Unimodal Feature Extraction.* We extract features from the three modalities as follows:

- Text: We concatenate the text of all the utterances and feed it into T5 [31] or Flan-T5 [6] to encode the representation of the text.
- Audio: We use the openSMILE toolkit to extract the acoustic features of each utterance.
- Vision: We apply the 3D-CNN network to obtain the visual features of each utterance, as mentioned in [35].

We treat the utterance representations from the audio and vision modalities as visual tokens, map them to the text representation space through a linear projection layer, and then concatenate them with textual tokens.

*5.1.3 Model Training.* We regard each emotion utterance and the preceding utterances within the sliding window as the context, and input them into the model through a carefully designed prompt template in the form of question answering: "*question: ... context: ...*". The instruction template for the question is "*What video caption suggests the emotion causes for [Rachel]'s [surprise] in U[3]?*", where the content in brackets is placeholder determined by samples. The context part contains the text, speaker, audio and video information of each utterance, such as "*U3. <A3> <V3> Rachel: Are you sure?*", where <A> and <V> respectively denote the representations of current audio and visual features after passing through a linear projection layer. The model is trained to generate the emotion-cause aware captions for cause generation by minimizing negative log-likelihood loss as follows:

$$\mathcal{L}_{vc} = -\sum_{i=1}^{m} \log P_\theta \left( w_i \mid w_{<i}, e, D \right), \quad (1)$$

where $e$ represents the given emotion and $D$ denotes the conversation context.

## 5.2 Multimodal Emotion Cause Generation

After obtaining the emotion-cause aware video caption predicted by our trained captioning model, we incorporate it into the input template as the supplement of the explicit audio-visual clues related to the given emotion: "*question: ... caption: ... context: ...*". The instruction question for cause generation is "*Why does [Rachel] feel [surprise] in U[3]?*". The context part fuses the multimodal information of the conversation in a similar way to the first step. Negative log-likelihood loss is also used as the optimization target, and the model is trained to generate the abstractive emotion cause.

$$\mathcal{L}_{ecg} = -\sum_{i=1}^{m} \log P_\theta \left( w_i \mid w_{<i}, e, D, C \right), \quad (2)$$

where $C$ denotes the predicted captions from the ECCap model.

## 6 EXPERIMENTS

### 6.1 Experiment Settings

*6.1.1 Evaluation Metrics.* For **automatic evaluation**, we report the following standard text generation metrics: BLEU [28], METEOR [1], ROUGE-L [25], and CIDEr [34] for lexical overlap, and BERTScore [40] for semantic similarity. The pre-trained model *deberta-xlarge-mnli* is used to obtain embeddings and compute the

Table 3: Performance Comparison of Different Methods on Our MECGC Task

| Modality | Method | BLEU-1 | BLEU-2 | BLEU-3 | BLEU-4 | METEOR | ROUGE-L | CIDEr | F_BERT |
|---|---|---|---|---|---|---|---|---|---|
| Text | GPT-3.5 (3-shot) | 0.2413 | 0.1582 | 0.1157 | 0.0886 | 0.1722 | 0.2258 | 0.8518 | 0.6801 |
| | Gemini (3-shot) | 0.2624 | 0.1726 | 0.1272 | 0.0984 | 0.1767 | 0.2497 | 0.8926 | 0.6936 |
| | T5 | 0.4815 | 0.4135 | 0.3718 | 0.3403 | 0.2979 | 0.4608 | 2.8770 | 0.7619 |
| | Flan-T5 | 0.4897 | 0.4212 | 0.3804 | 0.3499 | 0.3057 | 0.4751 | 3.0279 | 0.7698 |
| MM | Gemini-Pro (3-shot) | 0.2780 | 0.1826 | 0.1371 | 0.1085 | 0.1798 | 0.2453 | 0.8960 | 0.6960 |
| | ObG | **0.5011** | **0.4341** | **0.3939** | **0.3641** | 0.3008 | 0.4712 | 3.0079 | 0.7672 |
| | Flan-ObG | 0.4967 | 0.4313 | 0.3924 | 0.3631 | **0.3042** | **0.4781** | **3.0808** | **0.7711** |

F1 of BERTScore, i.e., F_BERT. We also further perform **human evaluation** on the captions and causes generated by our ObG framework. We randomly sample 100 conversations from the test set and employ an expert annotator to grade each prediction with a score 1-5 based on the following aspects: 1) fluency: whether it is grammatically correct and readable; 2) Coherence: whether it is coherent to the multimodal context; 3) Relevance: whether the predicted caption describes clues related to the cause or whether the predicted cause accurately describes what triggers the emotion.

*6.1.2 Implementation Details.* We evaluate our ObG framework on the ECGF dataset. ECGF is divided into training, validation, and test sets, the size of each is shown in Table 2. Both ECCap and CEM in our framework are initialized with pre-trained T5-base or Flan-T5-base. During fine-tuning, we used the AdamW optimizer with a weight decay of 0.01, and set the number of training epochs, batch size, and learning rate to 15, 16, and 1e-4, respectively. The context window range for CGM is set to [-5,2], i.e., the context between the five utterances preceding the target emotion utterance and the two utterances later, and the maximum input and output lengths are set to 512 and 40. To maintain consistency with the supervised data synthesis process, we set a smaller window range of [-3,0] for ECCap, and the maximum input and output lengths are 200 and 50 respectively. The results on the test set come from the best checkpoint regarding BLEU-4 on the validation set. We implement our framework with PyTorch and run all the experiments on an Nvidia RTX-3090 GPU.

## 6.2 Compared Methods

Since MECGC is a new task and there is no existing method for the task, we consider the following unimodal and multimodal methods as the comparison systems:

- GPT-3.5 [3] is a large language model developed by OpenAI. We apply GPT3.5 to emotion cause generation in a few-shot learning setting. Since it only takes text input, we concatenate the task prompt, three annotated samples, and the test conversation together and feed it, instructing it to output the cause for the given emotion. The specific task prompt is as follows: *In a conversation with multiple utterances, each including a speaker and the text, please write a sentence of no more than 40 words to describe what triggered the given emotion based on the context.*

[3]https://platform.openai.com/docs/models/gpt-3-5-turbo

Table 4: Human Evaluation of Generated Causes

| Methods | Fluency | Coherence | Relevance |
|---|---|---|---|
| Gemini-Pro (3-shot) | **4.93** | 3.65 | 3.47 |
| T5 | 4.82 | 4.05 | 3.35 |
| ObG | 4.86 | 3.84 | 3.52 |
| Flan-ObG | 4.90 | **4.21** | **3.59** |

- Gemini is a multimodal large language model developed by Google [33]. We also leverage its in-context learning capabilities to generate emotion causes. The text version, Gemini, uses the same instruction template as input to GPT-3.5. For the multimodal version, Gemini-Pro, we incorporate the keyframes from the video clips corresponding to each utterance.
- T5 [31] and Flan-T5 [6] are chosen as the backbones of our framework. We design the input template that includes the task prompt, caption and conversation context, and extend it to multimodal by incorporating representations of audio and vision modalities.

## 6.3 Main Results

*6.3.1 Automatic Evaluation Results.* In Table 3, we report the results of different methods on our MECGC task.

Firstly, we can observe that among all the text-based methods, GPT-3.5 (3-shot) performs relatively poorly while Gemini (3-shot) shows slightly better results. However, after performing instruction tuning on datasets, T5 and Flan-T5 significantly outperform both, especially in terms of BLEU-4 and CIDEr score, suggesting superior quality in emotion cause generation. Secondly, for the multimodal methods, Gemini-Pro (3-shot) performs slightly better than Gemini (3-shot), showing moderate results when introducing multimodal information. Both ObG and Flan-ObG show significant improvement, while ObG is particularly better in BLEU scores and Flan-ObG achieves higher performance on non-BLEU evaluation metrics. This could imply that ObG has become better at producing n-grams found in reference texts but may not have improved in terms of producing semantically accurate, fluent, and human-like text. This also underscores the importance of using a combination of metrics to evaluate text generation models comprehensively, as each metric captures different aspects of text quality.

**Table 5: Performance Comparison of T5-based Methods Across Different Modalities**

| T | A | V | Caption | BLEU-1 | BLEU-2 | BLEU-3 | BLEU-4 | METEOR | ROUGE-L | CIDEr | F_BERT |
|---|---|---|---------|--------|--------|--------|--------|--------|---------|-------|--------|
| ✓ |   |   | –        | 0.4815 | 0.4135 | 0.3718 | 0.3403 | 0.2979 | 0.4608 | 2.8770 | 0.7619 |
| ✓ |   | ✓ | –        | 0.4866 | 0.4200 | 0.3798 | 0.3492 | 0.3020 | 0.4657 | 2.9648 | 0.7661 |
| ✓ | ✓ | ✓ | –        | 0.4878 | 0.4202 | 0.3793 | 0.3483 | 0.2965 | 0.4620 | 2.9026 | 0.7625 |
| ✓ |   |   | Plain    | 0.4884 | 0.4189 | 0.3782 | 0.3483 | **0.3043** | 0.4666 | 2.9245 | **0.7685** |
| ✓ |   | ✓ | Plain    | 0.4796 | 0.4088 | 0.3670 | 0.3362 | 0.2977 | 0.4606 | 2.8725 | 0.7658 |
| ✓ | ✓ | ✓ | Plain    | 0.4803 | 0.4118 | 0.3717 | 0.3416 | 0.2976 | 0.4616 | 2.8925 | 0.7641 |
| ✓ |   |   | EC aware | 0.4811 | 0.4144 | 0.3751 | 0.3459 | 0.3001 | 0.4625 | 2.9373 | 0.7646 |
| ✓ |   | ✓ | EC aware | 0.4888 | 0.4214 | 0.3811 | 0.3508 | 0.3020 | 0.4648 | 2.9306 | 0.7667 |
| ✓ | ✓ | ✓ | EC aware | **0.5011** | **0.4341** | **0.3939** | **0.3641** | 0.3008 | **0.4712** | **3.0079** | 0.7672 |

**Table 6: Performance Comparsion of Pipeline and End-to-end framework**

| Method | B-4 | M. | R. | C. | F_B. |
|--------|-----|----|----|----|------|
| ObG | 0.3641 | 0.3008 | 0.4712 | 3.0079 | 0.7672 |
| ObG (E2E) | 0.3487 | 0.3006 | 0.4677 | 2.9551 | 0.7646 |
| Flan-ObG | 0.3631 | 0.3042 | 0.4781 | 3.0808 | 0.7711 |
| Flan-ObG (E2E) | 0.3507 | 0.3017 | 0.4679 | 2.9434 | 0.7668 |

*6.3.2 Human Evaluation Results.* We report the human evaluation results of causes generated by different methods in Table 4.

Firstly, it is easy to observe that Gemini-Pro (3-shot) receives the highest score for fluency but lower scores for coherence and relevance, indicating that while MLLM is able to generate fluent human-like text flow, it might not always be relevant to the context or the task. Secondly, both ObG and Flan-ObG show good performance in fluency, coherence, and relevance, especially Flan-ObG outperforms ObG in all metrics, indicating that Flan-ObG might better understand context and produce logical results when generating emotion causes. This observation is also consistent with the analysis from the automatic evaluation results.

## 6.4 Ablation Study on Modalities

In Table 5, we show the automatic evaluation metrics of integrating various types of captions across different modalities.

From the perspective of modalities, it can be observed that the vision modality plays a crucial role in generating emotion causes of higher quality, basically improving all evaluation metrics when jointly modeling with text modality. In some circumstances, integrating the audio modality information into the modeling process could result in the decrease of performance, which may be caused by the fact that simple audio processing in our experiment could introduce noise harmful for model performance. Notably, when we experiment with emotion-cause aware captions, jointly modeling text, audio and vision modality together lead to the best results for emotion cause generation, demonstrating that emotion-cause aware caption is able to compatible to different combination of modalities.

From the perspective of captions, introducing visual information in an image captioning manner can boost the model's performance

on multimodal emotion cause generation, especially when integrating the Emotion-cause aware captions into the model, achieving the highest score compared to no caption and plain caption.

## 6.5 In-depth Analysis

**Why choose pipeline framework instead of end-to-end framework?** In our experiments, we have attempted to implement our framework in an end-to-end manner, that is, using a single model optimized through multi-task learning to perform both video captioning and cause generation. The loss is the average of $\mathcal{L}_{vc}$ and $\mathcal{L}_{ecg}$. The comparison results of the pipeline framework and the end-to-end framework are shown in Table 6. It can be observed that, for both ObG and Flan-ObG, the pipeline framework performs better than end-to-end framework across all automatic evaluation metrics, indicating that the end-to-end framework is not always superior to the pipeline framework. Therefore, the pipeline framework is chosen as the main framework in this work.

**What is the quality of the generated captions?** We perform the human evaluation for the generated captions to access the their quality, and the evaluation results are shown in Table 8. We evaluate the quality from the three perspectives introduced in Table 4, i.e., fluency, coherence and relevance. Table 8 shows that Gemini-Pro (3-shot) receives the highest score for fluency, but lower scores for Coherence and Relevance. This suggests that while Gemini-Pro (3-shot) generates fluent and human-like text flow, it may not always be contextually relevant or coherent. Both ObG and Flan-ObG show better performance than Gemini-Pro (3-shot) in all three metrics, with Flan-ObG outperforming ObG in all metrics. This demonstrates that our ObG framework is able to generate more context and emotion cause captions compared to Gemini-Pro.

**What about generating caption on the utterance level?** As describe in Section 5.1.3, we perform conversation-level emotion-cause aware captioning. That is, for a conversation containing several utterances, we generate one caption by considering visual information from all the utterances. Another option is to perform utterance-level emotion-cause aware captioning, which generates captions for each utterance in a given conversation. We report the results of utterance-level captioning in Table 7. From Table 7, we can conclude that both ObG and Flan-ObG perform better with conversation-level captions, which could be due to that utterance-level captioning could introduce more noise from the utterance

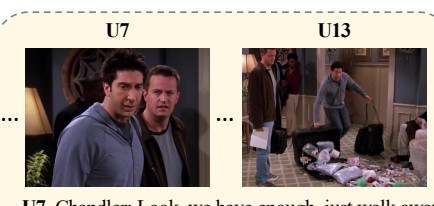

**U7**          **U13**

...          ...

**U7.** Chandler: Look, we have enough, just walk away.
**U8.** Ross: No, but I want... I want the pinecones!
**U9.** Chandler: There is a forest right outside.
**U10.** Ross: It is not the same.
**U11.** Chandler: Ok, go quick!
**U12.** Ross: Thank you for a delightful stay.
**U13.** Ross: Oh, my maple candy!

| Emotion | Annotation in ECF | Annotation in ECGF | Gemini-Pro (3shot) | Flan-T5 | Flan-ObG |
|---------|-------------------|--------------------|--------------------|---------|----------|
| U9 Chandler *anger* | U7, U8 | Ross wants the pinecones but there is a forest right outside. | Ross is taking too long to pick up the pine cones. | There is a forest right outside. | Ross wants the pinecones but there is a forest right outside. |
| U13 Ross *surprise* | U13 | Ross's maple candy dropped out. | Ross finds his maple candy. | Ross's maple candy tastes good. | Ross found his maple candy out. |

**ECCap:** A man struggles to carry a suitcase but all his belongings spills out of it.

**Figure 4: Case study on two representative test samples.**

**Table 7: Performance Comparison When Integrating Utterance-level Captions**

| Modality | Method | BLEU-1 | BLEU-2 | BLEU-3 | BLEU-4 | METEOR | ROUGE-L | CIDEr | F_BERT |
|----------|--------|--------|--------|--------|--------|--------|---------|-------|--------|
| Text | T5 | 0.4815 | 0.4135 | 0.3718 | 0.3403 | 0.2979 | 0.4608 | 2.8770 | 0.7619 |
| | Flan-T5 | 0.4897 | 0.4212 | 0.3804 | 0.3499 | **0.3057** | 0.4751 | 3.0279 | 0.7698 |
| MM | ObG-U | 0.4920 | 0.4237 | 0.3838 | 0.3544 | 0.2966 | 0.4612 | 2.9106 | 0.7625 |
| | Flan-ObG-U | 0.4927 | 0.4251 | 0.3846 | 0.3543 | 0.3034 | **0.4802** | 3.0573 | **0.7724** |
| | ObG | **0.5011** | **0.4341** | **0.3939** | **0.3641** | 0.3008 | 0.4712 | 3.0079 | 0.7672 |
| | Flan-ObG | 0.4967 | 0.4313 | 0.3924 | 0.3631 | 0.3042 | 0.4781 | **3.0808** | 0.7711 |

**Table 8: Human Evaluation of Generated Captions**

| Methods | Fluency | Coherence | Relevance |
|---------|---------|-----------|-----------|
| Gemini-Pro (3-shot) | **4.86** | 4.03 | 2.12 |
| ECCap | 4.70 | **4.31** | 2.15 |
| Flan-ECCap | 4.75 | 3.94 | **2.16** |

irrelevant to the given emotion cause, damaging the model performance. Consequently, we select the conversation-level strategy for emotion-cause aware captioning.

### 6.6 Case Study

We further conduct a case study to show the advantages of our MECGC task and our framework. As shown in Figure 4, we compare Flan-ObG with Gemini-Pro and Flan-T5 on two test samples from our dataset ECGF.

In the video clips of this conversation, Ross saw the pinecones when he was checking out and stuffed them in the suitcase before the receptionist arrived. The suitcase was too heavy to carry and fell open, revealing all the stuff. He found his maple sugar among the things that came out. The emotion causes for Chandler's *anger* in U9 are expressed in the text, but span two utterances. In our data set, the complete cause is described in one sentence. Our best model, Flan-ObG, accurately outputs the cause, while the text-only vanilla Flan-T5 only captures information from the target emotion utterance. Gemini seems to misunderstand the conversation text

and instead makes incorrect inferences, showing the hallucination phenomenon of LLM. On the other hand, Ross's *surprise* in U13 needs to be inferred by integrating the text and visual scene. Flan-T5 only takes text input, missing information from the visual modality, and therefore fails to learn the real cause. Gemini, having access to the visual modality, its output cause is roughly accurate. Based on the emotion-cause aware video caption generated by ECCap, our framework captures the key scene of "all his belongings spills out", generating a more relevant cause.

## 7 CONCLUSION

In this paper, we shift the focus of emotion cause analysis from traditional span or utterancce extraction to abstractive generation in multimodal conversations. Firstly, we introduced a new task named named Multimodal Emotion Cause Generation in Conversations (MECGC), aiming to directly generate a corresponding abstractive cause that summarizes all the clues triggering the emotion in the given utterance. Moreover, we constructed a new dataset named ECGF by manually annotating the abstractive causes for each emotion utterance in the existing ECF dataset. To address the new task, we further propose an "Observe-before-Generate" (ObG) framework, which integrates the multimodal information to generate the emotion-cause aware video caption, followed by incorporating the generated caption into the multimodal conversation for emotion cause generation. Experimental results on our dataset demonstrate the effectiveness of the proposed ObG framework and the usefulness of emotion-cause aware video captions.

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
