# OpenReview forum: "Observe before Generate: Emotion-Cause aware Video Caption for Multimodal Emotion Cause Generation in Conversations"
_acmmm.org/ACMMM/2024/Conference — MM2024 Poster_

### Official Review · Reviewer_ENRs · 2024-05-08

**Rating:** 4
**Confidence:** 4

**Summary:**

This paper introduces a new task, MECGC, which aims to directly obtain an abstractive cause that summarizes all the clues triggering the emotion in the utterance.  Position-oriented Prompt-tuning model for causal emotion entailment task. Furthermore, An "Observe-before-Generate" framework is constructed for this task, first generating emotion-cause-aware video captions based on multimodal information. Then, it merges the generated captions into multimodal dialogues to generate emotion causes.

**Strengths:**

1. The abstractive cause descriptions are easier to understand and subsequently process.
2. The model is simple and intuitive.

**Limitations:**

1. The paper's motivation is not clear, and it is essential to explain why multimodal and the abstract description.
2. The professionalism of the annotator in section 4.2.1 is suspicious. The emotion cause analysis requires expertise in the field and knowledge of psychology and linguistics, and it is suggested that experts in these fields evaluate the annotation results.
3. The performance improvement of Flan-ObG in Table 3 appears to be quite marginal, with only a 0.007 increase in BLEU-1, which is even smaller than the effect of random seed variation. This seems to reflect the lack of importance placed on audio-visual modal features. Furthermore, there is no mention of the stability and importance of multiple trials using different seeds to verify the performance improvement.
4. Section 6.3.2 mentions human evaluation, which involves evaluating Fluency, Coherence, and Relevance, but no specific evaluation method is given.
5. The correlations obtained in Table 4 are relatively low, and I believe the model has yet to obtain abstract descriptions of relatively high quality.
6. Although the paper contains many ablation experiments, the performance differences between the parts are not very significant.

**Suitability:**

3

---

### Official Review · Reviewer_sPmT · 2024-05-24

**Rating:** 6
**Confidence:** 4

**Summary:**

This paper introduces the task of Multimodal Emotion Cause Generation in Conversations (MECGC), aiming to generate summaries of emotion causes from multimodal contexts. A dataset called ECGF, with 1,374 conversations and 7,690 emotion instances from TV series, is created. The proposed framework uses video captions to improve the generation of emotion causes, demonstrating the significance of multimodal information in emotion cause analysis.

**Strengths:**

1. The study presents a new task, Multimodal Emotion Cause Generation in Conversations (MECGC), which focuses on creating abstractive summaries of emotion causes from multimodal data, pushing forward the boundaries of emotion cause analysis research.

2. The researchers developed a specialized dataset called ECGF by annotating abstract causes for each emotion within the existing ECF dataset, providing an enriched resource for investigating the MECGC task.

3. The paper introduces an innovative "Observe-before-Generate" (ObG) framework. This approach involves training an emotion-cause captioning model using examples generated by a Multimodal Large Language Model (MLLM), which then facilitates effective emotion cause generation from video captions.

Overall, this paper is the first to explore the multimodal emotion cause analysis task from a generative perspective, which is novel. The methods effectively integrate multimodal language models, providing meaningful results. The experimental analysis is comprehensive.

**Limitations:**

1. The conclusion section contains mixed tenses, which can confuse the reader and affect the overall coherence of the paper. The tenses should be consistent, preferably in the past tense, as the actions and results discussed have already been completed.
2. It is suggested to add some numbers for the questions and corresponding pairs in Section 6.5 to enhance clarity and readability.

**Suitability:**

3

---

### Official Review · Reviewer_xmTU · 2024-05-24

**Rating:** 4
**Confidence:** 3

**Summary:**

The paper introduces a new task and framework for analyzing emotions in conversations, specifically focusing on generating abstract summaries that describe the causes of emotions. The task is named Multimodal Emotion Cause Generation in Conversations (MECGC). The main contributions are the proposed dataset and benchmark.

**Strengths:**

- The paper is well structured and easy to follow.

- The experimental results appear to be promising.

- The constructed dataset is meaningful for the research field.

**Limitations:**

- It would be beneficial to include more references in the first paragraph of the Introduction.

- The dataset constructed by the authors is highly similar to ECF [35]; it would be advantageous to provide more intuitive examples for comparison in terms of abstraction and concretization.

- The specific version of GPT-3.5 used is not stated.

- The authors are advised to include results for GPT4v, or if possible, GPT4o.

**Suitability:**

3

---

### Official Review · Reviewer_S5p5 · 2024-05-30

**Rating:** 4
**Confidence:** 3

**Summary:**

The paper introduce a new task named Multimodal Emotion Cause Generation in Conversations (MECGC), which aims to generate an abstractive summary describing the causes that trigger the given emotion based on the multimodal context of conversations.
This work reconstructs a dataset based on ECF.

**Strengths:**

1.	This paper introduces a task named Multimodal Emotion Cause Generation in Conversations (MECGC), aiming to generate abstractive causes based on multimodal context.
2.	This paper constructs a dataset named ECGF for the MECGC task by annotating abstractive causes for each emotion based on the existing ECF datset.
3.	This paper proposes a multimodal pipeline framework named “Observebefore-Generate” (ObG), which first trains an ECCap model with examples synthesized by a MLLM to generate emotioncause aware video captions and then uses them for emotion cause generation.

**Limitations:**

1.	In line 206-214, the emotion cause analysis in conversations is first proposed by RECCON(https://arxiv.org/pdf/2012.11820 ), This work is very important in this scenario.
2.	Which utterance is the cause of the emotion that does not to be classified? Just need to generate a description of the emotional cause?
4.	Does the training process of ECCap use the whole annotated dataset?

**Suitability:**

2

---

### Meta-Review · Area_Chair_V2k9 · 2024-06-28

**Recommendation:** Accept (Poster)
**Confidence:** 5

**Metareview:**

The paper introduces a novel task called Multimodal Emotion Cause Generation in Conversations, which focuses on generating abstractive summaries describing the causes of emotions based on multimodal conversational contexts. The authors present a new dataset, ECGF, created by annotating abstractive causes for emotions within the existing ECF dataset. They propose an "Observe-before-Generate" framework that first uses an ECCap model to generate emotion-cause aware video captions and then employs these captions for emotion cause generation. The reviewers highlighted several strengths, including the introduction of a meaningful new task, the creation of a valuable dataset, and the innovative approach of integrating multimodal information for emotion cause analysis. The experimental results are promising, and the paper is well-structured and easy to follow. Some limitations were noted, such as the need for more detailed explanations of certain aspects and comparisons with existing datasets, as well as suggestions to include results from more advanced models like GPT-4. Overall, the authors have addressed the reviewers' concerns effectively during the rebuttal phase, and the paper presents significant contributions to the field, justifying its acceptance.